# Enhanced Photocatalytic Activity of Semiconductor Nanocomposites Doped with Ag Nanoclusters Under UV and Visible Light

**Jorge González-Rodríguez [1],\***, **Lucía Fernández [1]**, **Yanina B. Bava [2]**, **David Buceta [2]**,
**Carlos Vázquez-Vázquez [2]**, **Manuel Arturo López-Quintela [2]**, **Gumersindo Feijoo [1]** and
**Maria Teresa Moreira [1]**

[1] CRETUS Institute, Department of Chemical Engineering, Universidade de Santiago de Compostela, 15782 Santiago de Compostela, Spain; luciaf.fernandez@usc.es (L.F.); gumersindo.feijoo@usc.es (G.F.); maite.moreira@usc.es (M.T.M.)

[2] Laboratory of Magnetism and Nanotechnology, Department of Physical Chemistry, Faculty of Chemistry, Universidade de Santiago de Compostela, 15782 Santiago de Compostela, Spain; ybava@quimica.unlp.edu.ar (Y.B.B.); david.buceta@usc.es (D.B.); carlos.vazquez.vazquez@usc.es (C.V.-V.); malopez.quintela@usc.es (M.A.L.-Q.)

\* Correspondence: jorgegonzalez.rodriguez@usc.es; Tel.: +34-8818-16771

**Abstract:** Emerging contaminants (ECs) represent a wide range of compounds, whose complete elimination from wastewaters by conventional methods is not always guaranteed, posing human and environmental risks. Advanced oxidation processes (AOPs), based on the generation of highly oxidizing species, lead to the degradation of these ECs. In this context, $TiO_2$ and ZnO are the most widely used inorganic photocatalysts, mainly due to their low cost and wide availability. The addition of small amounts of nanoclusters may imply enhanced light absorption and an attenuation effect on the recombination rate of electron/hole pairs, resulting in improved photocatalytic activity. In this work, we propose the use of silver nanoclusters deposited on ZnO nanoparticles (ZnO–Ag), with a view to evaluating their catalytic activity under both ultraviolet A (UVA) and visible light, in order to reduce energetic requirements in prospective applications on a larger scale. The catalysts were produced and then characterized by scanning electron microscopy (SEM), X-ray diffractometry (XRD) and inductively coupled plasma-optical emission spectrometry (ICP-OES). As proof of concept of the capacity of photocatalysts doped with nanoclusters, experiments were carried out to remove the azo dye Orange II (OII). The results demonstrated the high photocatalytic efficiency achieved thanks to the incorporation of nanoclusters, especially evident in the experiments performed under white light.

**Keywords:** AOPs; zinc oxide; nanoclusters; photocatalysis; UVA; visible light

## 1. Introduction

Environmental awareness has identified water scarcity as a problem of increasing magnitude in many areas due to its decisive and essential role in life. The increase in world population, changes in consumption patterns, the high demand for water in intensive irrigation agriculture or the frequent events of floods and droughts leads to the depletion of many water resources and the unequal distribution of water in different regions of the planet [1]. In this context, the development of novel analytical methods for the analysis of water and the improvement of existing ones reveal the presence of ECs in both drinking water and wastewater effluents. Some of these compounds may be toxic to terrestrial and aquatic organisms at low concentrations [2]. These compounds represent a scientific-technological challenge, since the existing plants have not been designed for their elimination.

On the other hand, the regulation foreseen for the elimination of this type of contaminants implies that the new facilities must face their efficient removal [3,4]. For example, the release of wastewater with pharmaceutical products and bacteria caused the increased resistance to amoxicillin/clavulanic acid in *Salmonella enterica* strains from 1% to 7–16% within 15 years (2003–2018). Furthermore, other compounds such as endocrine disrupting chemicals (EDCs) can be released into the environment, stored in the organisms due to their recalcitrant properties and stability, and accumulated in certain organs producing long-term effects [5]. In view of the above, it is necessary to ensure treatment systems that are capable of eliminating ECs in the different water matrices due to the adverse effects on human health and ecosystem [6].

Thus, advanced oxidation processes (AOPs) represent an alternative to conventional methods to remove these contaminants. AOPs are physicochemical processes that involve the generation of reactive oxygen species (ROS), which are effective against oxidation of organic matter because they have high oxidation potentials capable of reacting and degrading a wide range of contaminants [7]. In recent years, the use of heterogeneous photocatalysts has been intensively studied for their wide application for environmental protection, with special attention to wastewater treatment [8]. Some authors have studied the degradation of the ECs using different types of catalyst based on metallic oxides such as ZnO or $TiO_2$ for the degradation of pharmaceuticals and personal care products (PPCPs) [9], pesticides [10] or industrial contaminants [7]. However, photocatalysis may be limited by costly energy requirements associated with the use of ultraviolet (UV) lamps, which also have limitations related to low quantum efficiency. Thus, photoactive materials are being developed, whose catalytic activity takes place in the optical window of visible light [11].

When semiconductor materials are used in photocatalysis, the photocatalyst is irradiated with light (h$\nu$) of equal or greater energy than its characteristic band gap, so that the electrons (e$^-$) of the valence band (VB) are promoted to the conduction band (CB), thereby generating electron/hole (e$^-$/h$^+$) pairs. Other mechanistic aspects are based on two broad types of simultaneously occurring photochemical reactions on the surface of the catalyst. The first involves reduction, from the photo-induced negative electrons at the CB to the dissolved $O_2$ present in the medium, producing superoxide ions ($O_2^{-\bullet}$), which can form hydroxyl radicals (OH$^\bullet$) and hydrogen peroxide ($H_2O_2$) in acidic medium. $H_2O_2$ may also decompose to OH$^\bullet$ under irradiation. The second involves oxidation, from the photoinduced positive holes at the VB, which react with $H_2O$ or hydroxyl ions (OH$^-$) to produce OH$^\bullet$. The active oxygen species $O_2^{\bullet-}$ and OH$^\bullet$, and h$^+$ react with organic molecules, triggering their consequent degradation (Figure 1) [12].

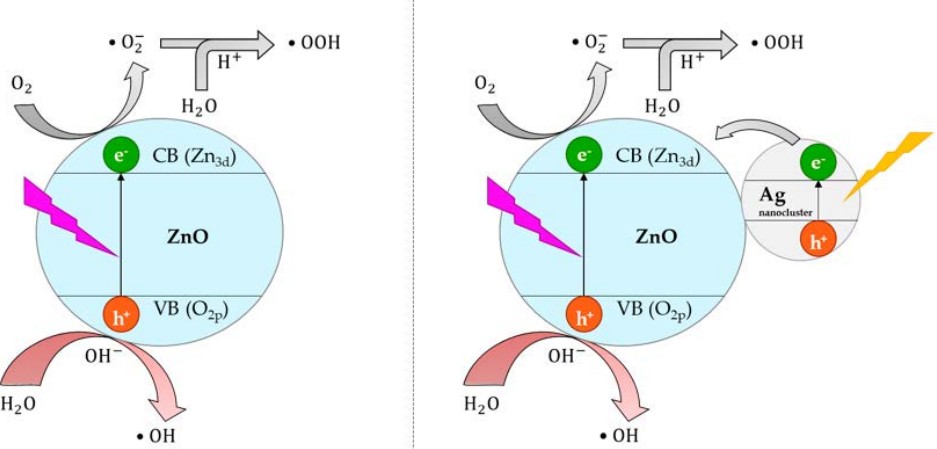

**Figure 1.** General mechanism of semiconductor photocatalysis (**left**) and modification of energy levels with the incorporation of silver nanoclusters (**right**).

Among the different photocatalysts, ZnO has received considerable attention due to its exceptional optoelectronic properties, strong oxidation capability, abundance and physicochemical stability [8]. In this context and with a view to their application in wastewater treatment, the photocatalytic efficiency of ZnO nanoparticles (NPs) has been evaluated in the degradation of pharmaceutical and personal care products (PPCPs) [13] or dyes [14].

However, the photocatalytic efficiency of ZnO NPs is often limited by an inefficient absorption of visible light and a rapid recombination rate of $e^-/h^+$ pairs. To avoid these disadvantages, the incorporation of metallic and non-metallic elements inside the crystal lattice of semiconductors reduces the recombination of electron-hole pairs. These compounds act by creating new energy levels between the CB and the VB that act as electron traps. Examples of doping with metallic silver or copper on ZnO or $TiO_2$ were reported to improve photocatalytic efficiencies [7,15–19].

On the other hand, these drawbacks can be circumvented also by depositing noble metals on the surface of the catalyst, which trap electrons of the CB of the semiconductor, thus reducing the possibilities of $e^-/h^+$ recombination and increasing the photocatalytic activity (Figure 1) [7]. In the search for novel materials capable of degrading organic pollutants in water under sunlight, ZnO NPs have been synthesized and functionalized with silver nanoclusters by a simple and green deposition method in water, conducted at ambient conditions.

Nanoclusters of metal elements are particles with low numbers of atoms, from 2 to ≤100 atoms, with sizes below ≈1.5–2 nm, and properties dramatically different from what would be expected from the scaling laws that govern the behavior of bulk and metal nanoparticles [20]. Nanoclusters of metal elements show the presence of discrete energy states and a sizable HOMO-LUMO bandgap, similar to the conduction band–valence band in semiconductors and lose the metallic behavior. This bandgap can be tuned by changing the number of atoms, the type of metal and the supporting material, and they can be used for different catalytic applications (heterogeneous catalysis, photocatalysis and electrocatalysis) [20–23].

In this article, the crystallinity, optical properties and morphology of the nanostructures obtained, ZnO–Ag, have been evaluated. Finally, the photocatalytic activity of ZnO–Ag nanocomposites with different Ag loadings has been studied in the removal of the dye Orange II (OII), used as model compound of organic pollutant, under UVA and white light.

## 2. Results and Discussion

### 2.1. Characterization of Catalysts

A sample containing silver nanoclusters of ≤10 atoms was used for the deposition onto the ZnO NPs. These small nanoclusters show planar geometries, as it can be shown by atomic force microscopy (Figure S1), confirming the presence of nanoclusters of ≤10 atoms [24].

Different Ag loads were applied on the surface of the ZnO NPs, so that four types of ZnO–Ag NCs with an Ag content of 1.3, 2.9, 3.2 and 7.4% (*w/w*) were obtained. The samples were structurally characterized by X-ray diffraction patterns. The two crystalline phases present in the samples were metallic silver (Ag, JCPDS PDF-2 card number 04-0783, peaks highlighted with red down-pointing triangles in Figure 2) and zincite (ZnO, JCPDS PDF-2 card number 36-1451, peaks highlighted with black up-pointing triangles in Figure 2) with hexagonal wurtzite structure.

No additional peaks were observed in the patterns, revealing the absence of impurity phases in the catalyst. Furthermore, there was no significant shift of the diffraction peaks, proving that silver atoms did not substitute any Zn sites in the lattice but were deposited onto the surface of ZnO. Figure 3 shows the morphology of the ZnO NPs (left) and ZnO–Ag NCs (right) observed by FE-SEM. ZnO NPs are present in the form of spherical aggregates of different sizes, between 50 and 500 nm. These aggregates are composed of smaller ZnO NPs (10–15 nm). In the case of the ZnO–Ag NCs, the presence of separate Ag nanoparticles along with the spherical aggregates is shown in Figure 3 (right).

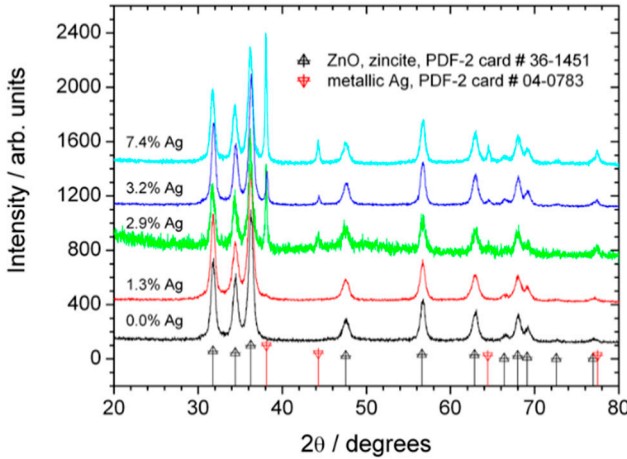

**Figure 2.** X-ray diffraction patterns of ZnO NPs and hybrid ZnO–Ag NCs. The main reflections from zincite (ZnO, JCPDS PDF-2 card number 36-1451) and metallic silver (Ag, JCPDS PDF-2 card number 04-0783) are included as drop lines.

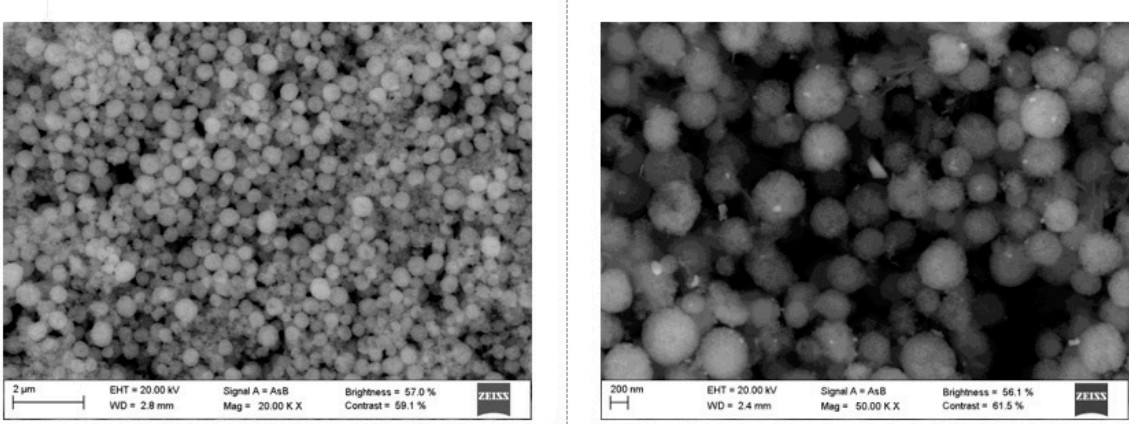

**Figure 3.** Scanning electron micrograph of the spherical ZnO aggregates (**left**); Detailed scanning electron micrograph of the ZnO–Ag nanocomposites, showing the presence of small Ag nanoparticles as brighter spots (**right**).

### 2.2. Influence of Ag Nanoclusters Loading onto Photocatalytic Activity of ZnO

Experiments were performed using fixed concentrations of a photocatalyst (200 mg L$^{-1}$) and OII (10 mg L$^{-1}$) in 10 mL of aqueous solutions, under UVA light for 60 min or white light for 180 min (Figure 4). In the absence of photocatalysts, photolysis controls resulted in OII degradation of 9% under UVA and 5% under white light, while adsorption studies of samples kept in dark conditions showed no OII removal (data not shown). When comparing the decolorization results, using ZnO nanoparticles as photocatalyst, no significant improvement in dye removal was observed, with maximum percentages of 16% and 9% under UVA and white light, respectively. As a general rule for all the experiments, OII degradation exhibited accelerated kinetic rates under UVA irradiation, which is attributed to a strong light absorption of these wavelengths by ZnO, while ZnO absorption in the visible region is weaker. It can be noted that the photocatalytic performance of the ZnO NPs improved with the addition of Ag nanoclusters, obtaining 97% and 49% of OII removal in the presence of ZnO–Ag with 1.3% (*w/w*) using UVA or white light, respectively. Photocatalytic performance gradually decreased with increasing Ag loads in the ZnO NPs. Therefore, there is evidence of the existence of an optimal silver loading to enhance the photocatalytic activity of the NC. This can be explained by the specific surface of ZnO available to interact with incident light, being lower with increasing concentrations of Ag nanoclusters in the NCs [25]. In fact, the decoration of the ZnO NPs with Ag nanoclusters leads

to a color modification, from white to brownish and grey, due to the formation of Ag nanoparticles, which improves the absorption of the ZnO–Ag NCs in the visible region (Figure 4).

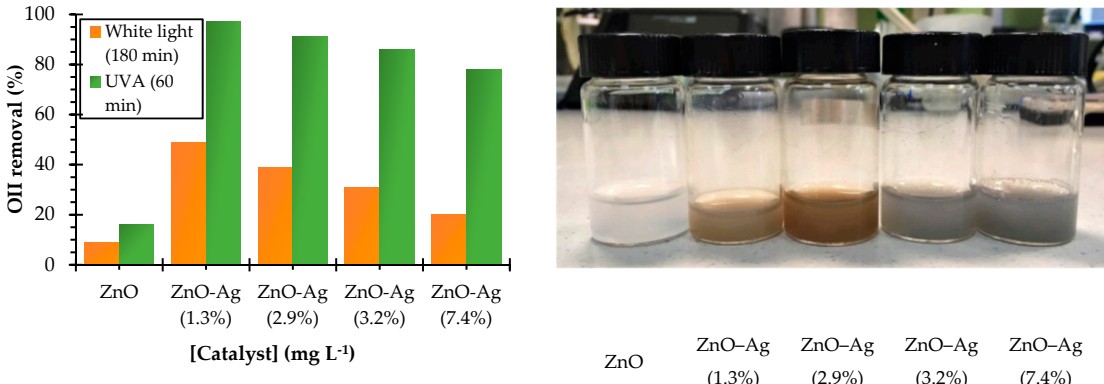

**Figure 4.** Silver loading effect on photocatalytic performance. The values in brackets correspond to the percentage of Ag in each NC (**left**); Aqueous suspensions of ZnO-NPs and ZnO–Ag NCs with different silver loadings (**right**).

However, according to previous findings, above the optimum Ag loading effectively deposited onto the NC, this can lead to an enhancement of the $e^-/h^+$ recombination rate, acting itself as a recombination center, thus contributing to a decrease in photocatalytic efficiency [26].

### 2.3. Influence of Photocatalyst Concentration on OII Removal

From the previous results, ZnO–Ag with 1.3% (*w/w*) of Ag was selected as the most effective photocatalyst. The effect of ZnO–Ag (1.3%) concentration was evaluated in 10 mL of aqueous samples containing an initial OII concentration ($C_{OII,i}$) of 10 mg $L^{-1}$, which were subjected to white light with 200–1000 mg $L^{-1}$ of ZnO NPs or ZnO–Ag (1.3%) NCs, and to UVA light with 50–500 mg $L^{-1}$ of photocatalyst (Figure 5). After 3 h of white light irradiation, samples containing from 200 to 750 mg $L^{-1}$ of ZnO–Ag (1.3%) showed increasing photocatalytic rates, obtaining 49 to 78% of OII elimination, respectively. This can be explained by the enhancement of the active sites in the catalyst by increasing their concentration in the samples.

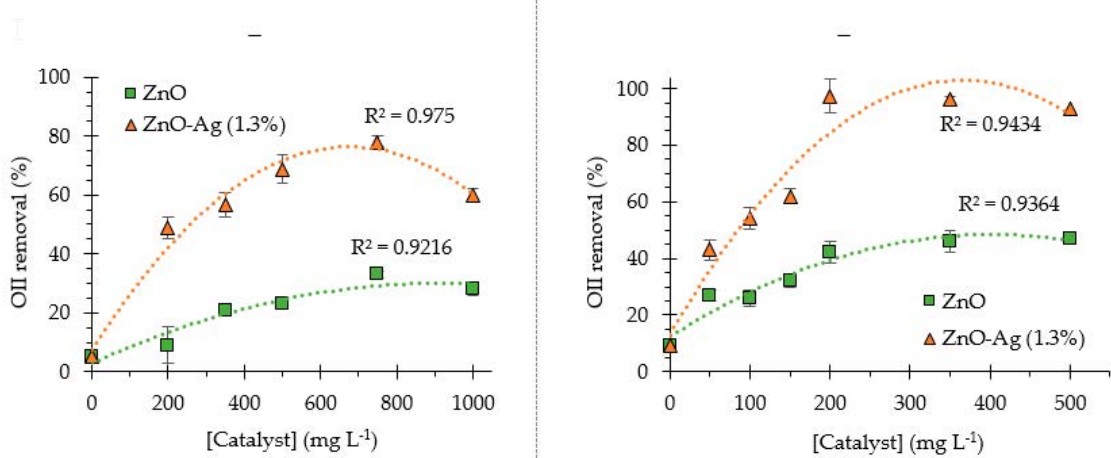

**Figure 5.** Influence of photocatalyst concentration under white light irradiation for 180 min (**left**) and UVA light irradiation for 60 min (**right**). Error bars were calculated considering a normal distribution, for $p < 0.01$ [obtained from kinetic data].

After 5 h of white light irradiation of the sample containing ZnO–Ag (1.3%) NCs at 750 mg $L^{-1}$, it was found that the OII concentration was below the detection limit of the spectrophotometric method, which was evidenced by the complete color removal of the sample. However, by increasing the concentration of the photocatalyst to a value of 1000 mg $L^{-1}$ of ZnO–Ag (1.3%), the extent of decolorization was inferior to that of the maximum: 60% OII removal. This phenomenon was observed by other authors for different photocatalysts [27,28] and is probably due to the effects of the NP aggregation and the reduction of the available surface area for photon absorption. A similar trend, but with slower reaction kinetics, was observed in samples containing ZnO NPs. As expected, the tests carried out under UVA light showed full removal of OII after 1 h, using ZnO–Ag (1.3%) NCs at 200 mg $L^{-1}$. These data were adjusted following a second order equation with an adequate fitting of data, achieving $R^2$ up to 0.92.

Accordingly, the negative natural logarithm of the ratio between OII concentration and its initial concentration, ln ($C_{OII,0}/C_{OII,t}$), was plotted as a function of the irradiation time and a linear regression was obtained. Correlation coefficients ($R^2$), half-lives ($t_{1/2}$) and apparent pseudo-first order rate constants (k) are presented in Table 1. $R^2$ ranges from 0.9323 to 0.9990, confirming the suitability of the pseudo-first order model to describe the kinetics of OII removal in the presence of ZnO–Ag (1.3%) and ZnO NPs, also applied by other authors to model the photocatalytic degradation of dyes and emerging contaminants. [16,27,29–31].

**Table 1.** Kinetic parameters for the photodegradation of OII with ZnO–Ag (1.3%) and ZnO NPs.

| Irradiation Source | [Catalyst] (mg $L^{-1}$) | ZnO–Ag (1.3%) | | | ZnO | | |
|---|---|---|---|---|---|---|---|
| | | $R^2$ | k ($h^1$) | $t_{1/2}$ (h) | $R^2$ | k ($h^{-1}$) | $t_{1/2}$ (h) |
| White light | 200 | 0.9861 | 0.235 ± 0.006 | 2.96 | 0.9425 | 0.036 ± 0.002 | 19.25 |
| | 350 | 0.9862 | 0.290 ± 0.007 | 2.41 | 0.9728 | 0.083 ± 0.003 | 8.25 |
| | 500 | 0.9930 | 0.390 ± 0.007 | 1.78 | 0.9847 | 0.089 ± 0.002 | 7.70 |
| | 750 | 0.9624 | 0.455 ± 0.017 | 1.52 | 0.9748 | 0.141 ± 0.005 | 4.81 |
| | 1000 | 0.9984 | 0.308 ± 0.003 | 2.27 | 0.9635 | 0.121 ± 0.004 | 5.78 |
| UVA light | 50 | 0.9859 | 0.541 ± 0.018 | 1.28 | 0.9323 | 0.346 ± 0.023 | 1.99 |
| | 100 | 0.9854 | 0.758 ± 0.026 | 0.92 | 0.9923 | 0.311 ± 0.008 | 2.22 |
| | 150 | 0.9823 | 1.006 ± 0.037 | 0.69 | 0.9988 | 0.388 ± 0.004 | 1.78 |
| | 200 | 0.9722 | 3.436 ± 0.176 | 0.20 | 0.9990 | 0.552 ± 0.005 | 1.26 |
| | 350 | 0.9778 | 3.019 ± 0.137 | 0.23 | 0.9951 | 0.633 [a] ± 0.012 | 1.10 |
| | 500 | 0.9946 | 2.650 ± 0.055 | 0.26 | 0.9925 | 0.642 [a] ± 0.016 | 1.08 |

[a] Not significantly different ($p < 0.05$).

Other works reported the photocatalytic degradation of OII using modified ZnO catalyst to improve kinetics. Chen et al. [27] obtained a kinetic constant of 0.033 $h^{-1}$ using micro-structured ZnO, which amounted to 0.065 $h^{-1}$ for ZnO decorated with Ag, both values lower than those obtained in these experiments, and they also used a larger catalyst concentration: 1500 mg $L^{-1}$. The enhancement of the results using nanostructured ZnO could take place because nanostructured catalysts have more surface/volume ratio than the micro-sized material, increasing the number of active sites per mass unit. Moreover, Siuleiman et al. [30] have structured ZnO in nanowires, obtaining kinetic constants of 0.092 and 0.112 $h^{-1}$ for UVA and visible light irradiation with a catalyst concentration of 500 mg $L^{-1}$, respectively.

In view of the above, the incorporation of the Ag nanoclusters causes an improvement of the kinetic constants of 3–6 times, both under UVA and white light radiation. In addition, the improvement in degradation rates by comparing the same catalyst concentrations under white and UVA light is 7–10 times greater. For all cases, the most notable differences occur for a catalyst concentration in the range of 200–500 mg $L^{-1}$. These ratios are similar to those obtained by Sornalingam et al. [31] using Au-$TiO_2$ NCs with UVA and cold white light. Reuse tests cannot be carried out due to the losses of catalyst at the recovery stage.

## 3. Materials and Methods

### 3.1. Materials

All chemicals used in this work were reagent-grade and were used without further purification. Diethylene glycol ($(HOCH_2CH_2)_2O$, DEG, 99%) was supplied by Alfa Aesar (Thermo Fisher, Kandel, Germany); Orange II ($C_{16}H_{10}N_2Na_2O_7S_2$, >85%), zinc(II) acetate dihydrate ($Zn(CH_3CO_2)_2 \cdot 2H_2O$, >98%), silver nitrate ($AgNO_3$, >99%) and absolute ethanol ($CH_3CH_2OH$, >99.8%) were supplied by Sigma-Aldrich (St. Louis, MI, USA). Silver nanoclusters (Ag-AQCs DS0481) were provided by NANOGAP SUB-NM POWDER, S.A (ZIP code 15895 O Milladoiro, A Coruña, Spain). This sample contains a mixture of Ag nanoclusters with ≤10 silver atoms per nanocluster (100 mg $L^{-1}$) and Ag(I) ions (400 mg $L^{-1}$).

### 3.2. Synthesis of Nanostructured Photocatalysts

### 3.2.1. ZnO Nanoparticles

The synthesis of ZnO NPs is based on the preparation of polyol-mediated ZnO [32]. In particular, 100 mL of 90 mM Zn(II) acetate solution in DEG were placed in a round-bottom flask and heated to 180 °C for 2 h under mechanical agitation. The obtained NPs were centrifuged at 7500 rpm for 15 min. Then, ZnO NPs were washed four times with ethanol. Finally, ZnO NPs were redispersed in water at a concentration ca. 0.83% (*w/w*) (determined by thermogravimetric analysis).

### 3.2.2. ZnO–Ag Nanocomposites

10 mL of the Ag nanoclusters stock solution (100 mg $L^{-1}$ in water) was placed in a 20 mL glass vial and pH was adjusted to 5 with $NH_4OH$ (28–30% *w/w*). To obtain nanocomposites (NCs) with different silver loadings, a given volume (1.1–4.1 mL) of the previous prepared stock solution of ZnO NPs (8.3 g $L^{-1}$) was added. The reaction mixture was incubated in an orbital shaker for 15 min (220 rpm, 24 °C). Then, the NC was centrifuged (7000 rpm, 25 min), the supernatant was removed, and the solids were re-dispersed in 20 mL of water. The sample was again centrifuged and the solids re-dispersed in 20 mL of water. Duplicate samples were prepared and after the first centrifugation step, the dispersion was subjected to a photochemical treatment at 254 nm for 15 min in order to reduce the residual Ag(I) ions present in the dispersion. The samples were then centrifuged and the solids re-dispersed in 20 mL of water. Blank samples of ZnO NPs without Ag nanoclusters were prepared following the same procedure but using 10 mL of an $AgNO_3$ solution (400 mg $L^{-1}$) to show the different behavior of clusters and Ag ions/nanoparticles formed in the blank.

### 3.3. Characterization of the ZnO–Ag Nanocomposites

The study of the crystalline phases was carried out by X-ray diffraction (XRD) in powder samples with a Philips PW1710 diffractometer (Cu Kα radiation source, λ = 1.54186 Å). Measurements were collected between 20° < 2θ < 80°, with steps of 0.020° and time per step of 5 s. The concentration of aqueous stocks of ZnO NPs was obtained by thermogravimetric analysis (TGA). The thermogravimetric curves were recorded with a Perkin Elmer TGA 7 thermobalance, operating under $N_2$ atmosphere, from room temperature to 850 °C, at a scanning rate of 10 °C $min^{-1}$.

Field-emission scanning electron micrographs were taken with a ZEISS FE-SEM ULTRA Plus microscope using the angle selective backscatter electron detector (AsB detector). The final concentrations of Zn and Ag were determined by inductively coupled plasma-optical emission spectrometry (ICP-OES) using a Perkin Elmer Model Optima 3300 DV spectrometer, equipped with an AS91 autosampler.

Atomic force microscopy (AFM) measurements were conducted under normal ambient conditions using an XE-100 instrument (Park Systems, Suwon, Korea) in non-contact mode. The AFM tips were aluminum-coated silicon ACTA from Park Systems with a resonance frequency of 325 kHz. For AFM

imaging, a drop of the Ag nanoclusters diluted sample was deposited onto a freshly cleaved mica sheet (Grade V-1 Muscovite) (Park Systems, Suwon, Korea), which was thoroughly washed with Milli-Q water and dried under nitrogen flow.

### 3.4. Photocatalytic Degradation of Orange II under UVA and White Light

The photocatalytic activities of ZnO and ZnO–Ag were evaluated by exposure of samples in glass beakers to UVA (365 nm wavelength UVP pen Ray model 11SC-1L) or white irradiation (fluorescent lamp PL G23 11 W 6400 K, 400–730 nm). Photocatalytic tests were performed on 10 mL of aqueous samples containing 50–1000 mg $L^{-1}$ of photocatalyst and 10 mg $L^{-1}$ of OII, at pH 7 and room temperature. The fluorescent lamp is located externally on one side, at approximately 3 cm from the vial, and the UVA lamp is in the center of the samples, using a submerged quartz tube. Two types of control samples were performed in parallel: direct photolysis control samples of OII under the same irradiation conditions; and adsorption control samples containing photocatalysis and OII, under the same sample preparation but kept in darkness. The solutions were stirred for 30 min in dark to achieve adsorption equilibrium. At regular intervals, spectrophotometric measurements were performed to monitor OII concentration in a BioTek PowerWave XS2 micro-plate spectrophotometer (Winooski, VT, USA). The photodegradation yield (%) was determined using the following equation:

$$\text{Yield (\%)} = (C_{OII,I} - C_{OII,t})/C_{OII,0} \times 100\% \tag{1}$$

### 3.5. Determination of Kinetic Parameters

The determination of kinetic parameters was performed by adjusting a pseudo-first order kinetic model (Equation (2)) to each set of photocatalyst concentration used in the UVA and white light studies. The linearization of this equation (Equation (3)) and the expression used to calculate the half-life (Equation (4)) are shown below:

$$C_{OII,t} = C_{OII,i}\ e^{-kt}, \tag{2}$$

$$\ln (C_{OII,i}/C_{OII,t}) = kt, \tag{3}$$

$$t_{1/2} = \ln(2)/k, \tag{4}$$

being k the kinetic constant; t the time of the experiment and $t_{1/2}$ the half-life of the compound under study.

## 4. Conclusions

ZnO nanoparticles were prepared by a simple polyol-mediated method and successfully decorated with Ag nanoclusters, obtaining a novel nanocomposite (ZnO–Ag) with different degrees of silver loadings (1.3–7.4% *w/w*). In addition, the final dispersions of nanoparticles received a photochemical treatment to remove the residual Ag, avoiding the interferences in the subsequent photodegradation step. The influence of the Ag content on ZnO regarding the removal of Orange II was studied, obtaining that the presence of this noble metal at 1.3% greatly enhanced the photocatalytic activity, which suggests the potential of this nanocomposite to be applied in prospective applications in the field of water treatment, both in drinking and wastewater treatment plants. Semiconductor photocatalysis represents a promising alternative to conventional technologies since the use of chemicals would be avoided and solar energy could be used as photon source. In addition, the unspecific oxidation mechanisms in AOPs allow degradation and mineralization of a wide range of pollutants.

In this work, photocatalytic studies were performed under UVA and white light, obtaining the optimum concentrations of catalyst and nanoclusters that achieved removal percentages up to 75% for visible light after 3 h and nearly complete removal for UVA after 1 h. Further research is needed to fully explore this photocatalysis in practical applications. One of the main drawbacks of this catalyst is its separation from the water matrix after treatment. Its immobilization on a suitable support that

avoids additional steps such as centrifugation, which is the method used so far, will allow the reuse of the photocatalyst in different water treatment cycles, bringing this research closer to a real wastewater treatment plant. An alternative for this immobilization is the deposition of the NPs onto magnetic nanoparticles, which can be easily separated by applying a magnetic field. Moreover, immobilization over supports such as silica, zeolites or alumina would improve the recovery of the catalysts towards their industrial applications.

**Supplementary Materials:** The following are available online at http://www.mdpi.com/2073-4344/10/1/31/s1. Figure S1: AFM topography image and line profiles of small Ag nanoclusters deposited on mica.

**Author Contributions:** Conceptualization and supervision, M.T.M., G.F., M.A.L.-Q. and C.V.-V.; investigation, J.G.-R.; resources, Y.B.B., D.B.; writing–original draft, L.F.; writing–review & editing, J.G.-R., M.T.M., C.V.-V. All authors have read and agreed to the published version of the manuscript.

**Funding:** This research was supported by two projects granted by Spanish Ministry of Science, Innovation and Universities: MODENA Project CTQ2016-79461-R and CLUSTERCAT Project MAT2015-67458-P, and Fundación Ramón Areces, Spain (Project CIVP18A3940).

**Acknowledgments:** J.G.-R. thanks the Xunta de Galicia Counseling of Education, Universities and Vocational Training for his predoctoral fellowship. M.T.M., G.F. and J.G.-R. belong to CRETUS Institute. The authors also thank Xunta de Galicia for the CRETUS (AGRUP2015/02) and AEMAT (ED431E-2018/08) Strategic Partnerships, and the use of RIAIDT-USC analytical facilities. The authors belong to the Galician Competitive Research Groups ED431C-2017/22 and ED431C-2017/29, co-funded by FEDER.

**Conflicts of Interest:** The authors declare no conflict of interest.

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
