# Peer review of "Enhanced Photocatalytic Activity of Semiconductor Nanocomposites Doped with Ag Nanoclusters Under UV and Visible Light"

_catalysts, doi:10.3390/catal10010031_

Round 1

Reviewer 1 Report

Please refer to enclosed file

Reviewer 2 Report

The paper reports the preparation and characterization of ZnO-Ag nanocomposites, as well as, the degradation of the dye Orange II under UVA and white light I suggest the paper be accepted for publication in “catalysts” Journal.

In the “Conclusions” section the authors wrote: “ZnO nanoparticles were prepared by a simple polyol-mediated method and successfully decorated with Ag atomic quantum clusters (Ag AQCs), obtaining a novel nanocomposite (ZnO-Ag) with different degrees of silver loadings (1.3-7.4% w/w).”, I think that the novelty is also and in the procedure, namely, the combination of the polyol mediated method and of the Ag AQCs photochemical treatment in order to reduce the residual Ag present in the dispersion.

Round 2

Reviewer 1 Report

The authors addressed all raised concerns and improved the manuscript according to the reviewer's suggestions. This revised version is worth for publication on Catalysts